# KinFormer: Generalizable Dynamical Symbolic Regression for catalytic organic Reaction Kinetics

**Jindou Chen**[†,1] **Jidong Tian**[†,1,2] **Liang Wu**[‡] **Xinwei Chen**[‡] **Xiaokang Yang**[†] **Yaohui Jin**[†,2] **Yanyan Xu**[†,2]
[†]MoE Key Lab of Artificial Intelligence, Shanghai Jiao Tong University
[‡]School of Chemistry and Chemical Engineering, Shanghai Jiao Tong University

## Abstract

Modeling kinetic equations is essential for understanding the mechanisms of organic chemical reactions, yet a complex and time-consuming task. Kinetic equation prediction is formulated as a problem of dynamical symbolic regression (DSR) subject to physical chemistry constraints. Deep learning (DL) holds the potential to capture reaction patterns and predict kinetic equations from data of chemical species, effectively avoiding empirical bias and improving efficiency compared with traditional analytical methods. Despite numerous studies focusing on DSR and the introduction of Transformers to predict ordinary differential equations, the corresponding models lack generalization abilities across diverse categories of reactions. In this study, we propose KinFormer, a generalizable kinetic equation prediction model. KinFormer utilizes a conditional Transformer to model DSR under physical constraints and employs Monte Carlo Tree Search to apply the model to new types of reactions. Experimental results on 20 types of organic reactions demonstrate that KinFormer not only outperforms classical SR algorithm baselines, but also exceeds Transformer baselines in out-of-domain evaluations, thereby proving its generalization ability.

## 1 Introduction

A mechanistic comprehension of chemical reactions is imperative for devising novel catalysts, exploring various modes of reactivity, and developing environmentally friendly and sustainable chemical processes (van Dijk et al., 2021; Salazar et al., 2020; Butcher et al., 2020; Hutchinson et al., 2021). Reaction mechanism and its kinetics are two sides of a chemical reaction, both of which are of fundamentally importance. Reaction mechanism helps us to gain insights into chemical reaction at a molecule level and identify the reaction pathway, and the reaction kinetics on the other hand is the quantitative understanding of the factors influencing the reaction rate. Conventional pipeline for kinetics analysis involves three stages: (1) assuming the possible reaction mechanism based on existing knowledge of chemistry; (2) deriving kinetic equations in differential form based on the possible mechanism and integrating the kinetic equations; (3) fitting the kinetic equations with experimental data (Bédard et al., 2018; Shi et al., 2021). However, traditional kinetics analysis is limited by human knowledge of chemical reaction and more probably introduces empirical errors, and requires case-by-case analysis, resulting in low efficiency and poor generalization (Burés & Larrosa, 2023). To avoid these challenges, recent studies of deep learning provide insights for reaction kinetics analysis, such as automatic reaction mechanism exploration (Yang et al., 2019; Jorner et al., 2021; Feng & Wang, 2023), kinetic property regression (Farrar & Grayson, 2022; García-Andrade et al., 2023) and rate constant estimation (Gao et al., 2016; Maeda et al., 2022). These methods typically target only one specific reaction or may require expert-guided training, and cannot be generalized.

Actually, the essence of reaction kinetics analysis lies in kinetic equation prediction (KEP), which combines mechanism generation and constant estimation. Specifically, a catalytic organic reaction of $S \xrightarrow{cat} P$ is taken as the example: $S$ (substrate) combines with $cat$ (catalyst) to form $catS$

---

[1]Equal contribution
[2]Corresponding authors: *{frank92, jinyh, yanyanxu}@sjtu.edu.cn*

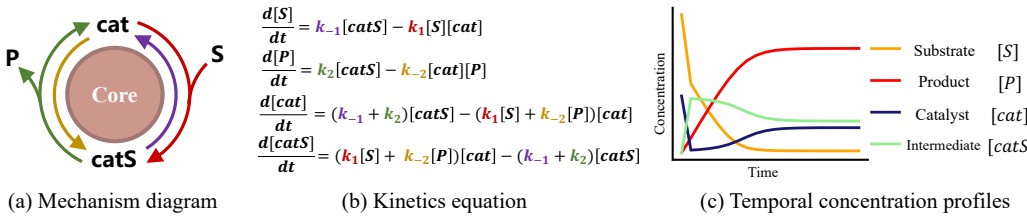

(a) Mechanism diagram         (b) Kinetics equation         (c) Temporal concentration profiles

Figure 1: Illustrations of catalytic organic reaction. (a) shows the core mechanism (M1) with transformation relations among species. (b) provides the governing equations (ODEs) under the constraints of mass-action law. (c) presents one of the corresponding temporal concentration profiles.

(intermediate), subsequently it decomposes to $P$ (product), and $cat$ is recovered, as shown in Figure 1a. According to the mass-action law, the chemical mechanism is mathematically controlled by corresponding ordinary differential equations (ODEs), shown in Figure 1b ($[\cdot]$ represents the concentration and $k$ is the reaction rate constant). The form of the ODEs is determined by the type of reaction mechanism. In this contribution, due to the dependence of kinetic data with the underlying reaction mechanism, the objective of KEP is to predict the whole set of ODEs from a group of kinetic data (temporal concentration profiles, shown in Figure 1c).

In this work, we innovatively formulate KEP as a dynamical symbolic regression (DSR) problem (Brunton et al., 2016; Weilbach et al., 2021) with physical constraints derived from the mass-action law. Our goal is to instruct model to successfully regress ODEs from temporal concentration profiles, and possess generalization capability. Classical SR methods (Brunton et al., 2016; Cranmer, 2023) are usually individual methods without any generalization ability, and performs poorly in specific domain due to a lack of pre-training stage. With advent of Transformer (Vaswani et al., 2017), researchers have developed pre-trained sequence-to-sequence Transformer models for SR tasks. ODEFormer (d'Ascoli et al., 2023) first introduces Transformer into DSR, showing surprising performance. However, directly applying ODEFormer to KEP faces challenges in generalization, as the common training strategies cannot make the model effectively learn physical constraints controlled by the mass-action law. On one hand, the **universal** training strategy (d'Ascoli et al., 2023) ODEFormer adopts, generates the entire ODEs in an end-to-end manner, leading to overfitting on training patterns, good in-domain performance, but poor generalization. On the other hand, inspired by P-tuning (Liu et al., 2022), the **independent** strategy generates each equation with the pre-defined prompt, completely ignoring the physical constraints and correlations among equations. To address these issues, we first design a **conditional** strategy to train Transformer on simulated kinetic data, which randomly selects several equations as conditions, and chooses an additional one to predict. The strategy adopted in the equation level is more balanced to enables the model to capture such physical correlations. However, **conditional** strategy requires the reasonable and explainable generation order to further improve the generalization ability of the model. To tackle this, we introduces a generation order search module with Monte Carlo Tree Search (MCTS), which combines with the conditional strategy and forms a new generalizable KEP framework, KinFormer. We experiment on a simulated dataset with 20 types of catalytic organic reactions (Burés & Larrosa, 2023). Results demonstrate the powerful generalization of KinFormer compared with other baselines. In addition, Monte Carlo trees provide an explanatory approach for how KinFormer predicts reaction kinetic equations.

In general, our contributions include: (1) We propose a generalizable training strategy, the conditional strategy, to solve improve the generalization ability of KEP. (2) We design an MCTS post-processing module to search the optimal generation order and providing explanations corresponding to the conditional strategy, which forms the KinFormer framework. (3) Experiments on a catalytic organic reaction simulated dataset demonstrate KinFormer's generalization ability.

## 2 RELATED WORKS

### 2.1 SYMBOLIC REGRESSION

Symbolic regression (SR) aims to discover the most optimal mathematical function that can accurately fit dataset. One of the dominant methods for SR is Genetic Programming (GP) algorithm that

simulates the evolution of human history (Schmidt & Lipson, 2009; Błądek & Krawiec, 2022; Trujillo et al., 2016; Tohme et al., 2022; Virgolin et al., 2021). Afterwards, SR has garnered increasing attention from the deep learning (DL) community, driven by the observation that neural networks (NNs) excel in discerning qualitative patterns (Petersen et al., 2019; Udrescu & Tegmark, 2020; Udrescu et al., 2020; Holt et al., 2023). The advent of transformer (Vaswani et al., 2017) have shown remarkable performance in natural language processing, which has inspired researchers to develop pre-trained transformer models for SR (Vastl et al., 2024; Lample & Charton, 2019; Charton, 2021). After large-scale pre-training on synthetic data, inference often achieves significant acceleration, since it necessitates no training for unseen dataset (Biggio et al., 2021). For example, the E2E framework (Kamienny et al., 2022) is end-to-end and designed to forecast the entire equation, including constants. Most of these works concentrate on functional SR, instead of ODEs.

## 2.2 DYNAMIC SYMBOLICAL REGRESSION

Dynamical SR, as a special case of SR, refers to the inference of ODEs based on temporal data $(t, x(t))$. A significant obstacle lies in the lack of regression targets $\dot{x}(t)$ since temporal derivatives are usually not observable directly. A typical remedy involves utilizing numerical approximations of the absent derivatives as substitute objectives (Gaucel et al., 2014; La Cava et al., 2016; Brunton et al., 2016). NNs have also been integrated with GP for dynamical SR (Udrescu et al., 2020; Omejc et al., 2023; Weilbach et al., 2021), but there still exists generalization problems due to the absence of prior knowledge. NSODE (Becker et al., 2023) is a transformer-based method for dynamical SR but can only work on univariate ODEs. ODEFormer (d'Ascoli et al., 2023) is proposed to infer multidimensional ODEs in symbolic form from the observation of a single solution trajectory. However, the training strategy of ODEFormer leads to poor generalization, especially in KEP.

## 3 PRELIMINARIES

### 3.1 KINETIC EQUATION PREDICTION

Kinetic equation prediction (KEP) is a kinetics analysis task to discover the ODEs (denoted by $\mathbf{f}$) of an catalytic organic reaction ($S \xrightarrow{cat} P$) according to temporal concentration profiles. Therefore, the input of KEP is the representation of profiles, a time series of $X = \{(t, \mathbf{x}(t)) | t \in \{1, 2, ..., T\}\}$, where $T$ is the maximum timestep and $\mathbf{x}$ is a $D$-dimensional vector corresponding to the concentrations of reaction species, including substrates, products, catalysts and intermediates ($\mathbf{x}_t = \{x^s(t) | s \in \{S, P, cat, catS, \cdots\}$). The profiles are governed by kinetic equations (ODEs), shown in Equation 1. The objective of KEP is to formulate $\mathbf{f}$ from observed temporal concentration profiles.

$$\frac{\mathbf{d}}{\mathbf{d}t}\mathbf{x}(t) = \mathbf{f}(\mathbf{x}(t)), \quad \mathbf{f} : \mathbb{R}^D \to \mathbb{R}^D \tag{1}$$

Hence, KEP can be formulated as a DSR task (Brunton et al., 2016; d'Ascoli et al., 2023), where $\mathbf{f}$ can be inferred in symbolic form constrained by the mass-action law $R$. In practice, $\mathbf{f}$ can be converted into prefix notations $\mathbf{y}$ for deep learning (Lample & Charton, 2019). It means that $\mathbf{y} = g(\mathbf{f})$ and $\mathbf{f} = g^{-1}(\mathbf{y})$ hold, where $g$ and $g^{-1}$ are conversion functions. Therefore, given the training dataset $\mathbf{D} = \{(X, \mathbf{y})\} = \{(t, \mathbf{x}(t), \mathbf{y})\}$ ($X = \{(t, \mathbf{x}(t)) | t \in \{1, 2, ..., T\}\}$, $\mathbf{y} = [y_1, y_2, \cdots, y_n]$, $y_i$ is the symbolic token in the ODE, such as "*add*", "*mul*" and discrete numeric symbols), deep learning-based KEP is to train a DSR NN $F_\theta$ with parameters $\theta$, making $\forall (X, y) \in \mathbf{D}$ satisfy Equation 2.

$$\begin{cases} \mathbf{y} = F_\theta(t, \mathbf{x}(t)), \quad \mathbf{f} = g^{-1}(\mathbf{y}) \\ \frac{\mathbf{d}}{\mathbf{d}t}\mathbf{x}(t) = \mathbf{f}(\mathbf{x}(t)), \quad R \vDash \mathbf{f} \end{cases} \tag{2}$$

$R \vDash \mathbf{f}$ means the form of $\mathbf{f}$ should satisfy physical constraints $R$. Actually, it is challenging to explicitly model $R$ into $F_\theta$ to guarantee compliance and generalization. In this work, we instead model $R$ implicitly through a novel training strategy and MCTS, which is consistent with Equation 2.

### 3.2 CATALYTIC ORGANIC REACTION

Although Equation 2 provides a general objective of KEP, we focus on catalytic organic reactions, following the previous Nature work (Burés & Larrosa, 2023). We have considered 20 commonly

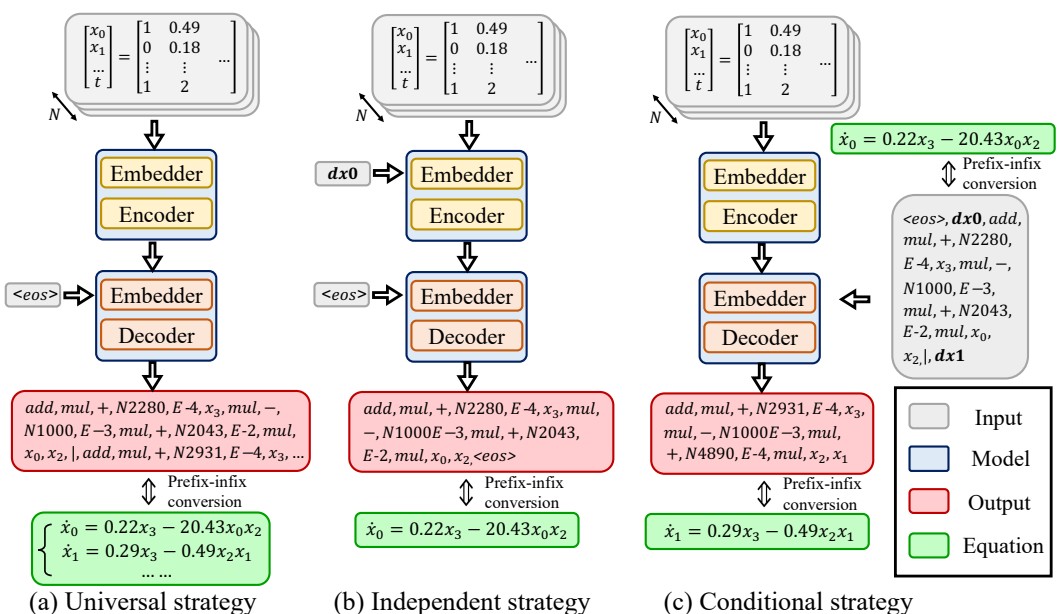

Figure 2: Comparison of three types of training strategies. (a) the framework of universal Transformer that generates the whole ODEs according a prior order. (b) the independent Transformer that independently generate each ODE through an additional prompt, such as **dx0**. (c) our proposed conditional strategy that provides contexts for decoder to capture correlations among different ODEs.

encountered types of catalytic organic reactions, whose species can be simplified into unified expressions: substrates ($S$), products ($P$), catalysts ($cat$) and intermediates ($catS, cat2S, \cdots$). These 20 types can further be categorized into four groups of distinct chemical mechanisms: (1) core mechanism (M1), as shown in Figure 1; (2) bicatalytic mechanisms (M2-M5); (3) catalyst activation mechanisms (M6-M8); (4) catalyst deactivation mechanisms (M9-M20) (See Appendix for more details). Like other chemical reactions, each type of reaction is mathematically described by a set of rigorously expert-verified ODEs function of kinetic constants $\mathbf{k}$ and concentration variable of chemical species. Therefore, solving ODEs allow the generation of an infinite number of temporal concentration profiles of reactions, constituting a kinetic space. In this work, we choose to solve the set of ODEs that describe the kinetic behavior of each mechanism as function of kinetic constants to create catalytic organic kinetic datasets.

## 4 METHODOLOGY

### 4.1 CONDITIONAL STRATEGY

Inspired by ODEFormer, we adopt the same Transformer backbone that encodes numerical inputs by the discrete three-token method (d'Ascoli et al., 2023). The model architecture, optimization and other encoding/decoding techniques are all aligned with ODEFormer. On the backbone, we explore three types of training strategies: universal, independent and conditional strategies.

**Universal strategy** provides an end-to-end encoder-decoder model that generates the whole ODEs for a reaction, which is also the strategy of ODEFormer. Figure 2a shows the whole framework of the universal one. The encoder is used to encode the discrete time series $X$, while the decoder generates each token autoregressively starting from the "$<eos>$" token. Therefore, the objective of the universal strategy is shown in Equation 3, where $y_0$ is the "$<eos>$" token, $\mathbf{y}_{<i} = [y_0, \cdots, y_{i-1}]$, $P_\theta(\cdot)$ is the probability calculated by the netwrok, and $\theta$ represents trainable parameters.

$$\max_\theta \log P_\theta(\mathbf{y}|X) = \log \prod_{i=1}^{n} P_\theta(y_i|X, \mathbf{y}_{<i}) = \sum_{i=1}^{n} \log P_\theta(y_i|X, \mathbf{y}_{<i}) \qquad (3)$$

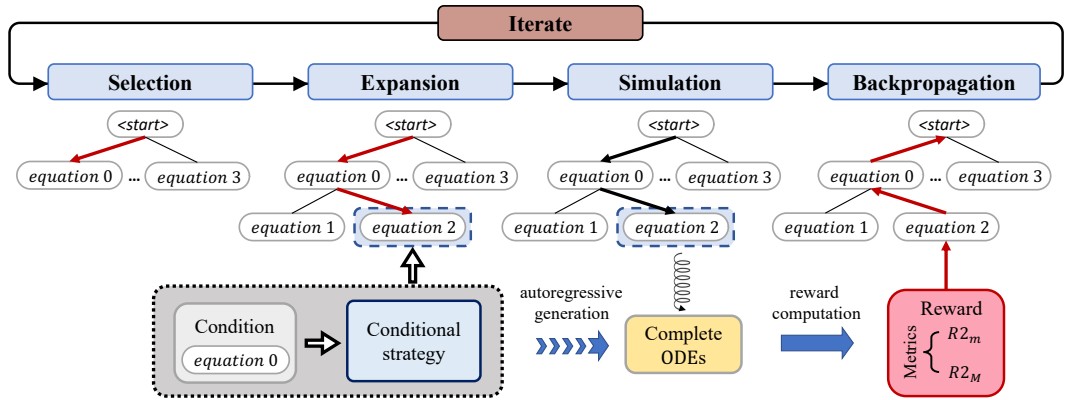

Figure 3: MCTS generation order search with based on conditional model.

**Independent strategy** re-splits the whole ODEs into each independent ODE. Specifically, given kinetic equations $\mathbf{y}$, it includes several separate ODEs $\mathbf{e}$. Therefore, $\mathbf{y} = [\mathbf{e}_1, \cdots, \mathbf{e}_D]$ holds, where $D$ is the dimension of species. $\mathbf{e}_i$ is a continuous token segment of $\mathbf{y}$ and represents the prefix notation of the $i_{th}$ ODE. To indicate which ODE to be predicted, the independent model introduce a prompt token $p_i$ with the form of "$\mathbf{dx}i$". Figure 2b shows the framework. Following P-tuning (Liu et al., 2022), the prompt is encoded in the encoder instead of the decoder. In general, the objective of the independent strategy is shown in Equation 4. From the equation, this strategy assumes the independence among ODEs, so each ODE is predicted separately through prompts.

$$\max_\theta \; \log \; P_\theta(\mathbf{y}|X) = \log \prod_{i=1}^{D} P_\theta(\mathbf{e}_i|X, p_i) = \sum_{i=1}^{D} \log \; P_\theta(\mathbf{e}_i|X, p_i) \tag{4}$$

**Conditional strategy** is a balanced strategy that we have proposed. It also re-splits $\mathbf{y}$ into ODE segment $\mathbf{e}_i$. Different from the independent one, conditional strategy models the correlations among $\mathbf{e}_i$ by introducing the condition $\mathbf{c}$ (Figure 2c demonstrate how to use the condition to prompt the model). To calculate unordered $\mathbf{c}$, we shuffle $[\mathbf{e}_1, \cdots, \mathbf{e}_D]$ and acquire a shuffled list $[\mathbf{e}'_1, \cdots, \mathbf{e}'_D]$, where $\mathbf{y} = \mathbf{shuffle}([\mathbf{e}_1, \cdots, \mathbf{e}_D]) = [\mathbf{e}'_1, \cdots, \mathbf{e}'_D]$. Therefore, $\mathbf{c}_i$ is comprised of the first $i$-1 ODEs from the shuffled list ($\mathbf{c}_i = [\mathbf{e}'_1, \cdots, \mathbf{e}'_{i-1}]$) and the objective of the conditional strategy is shown in Equation 5, where $\mathbf{c}_1$ is an empty set. From the equation, this strategy treats each ODE as a whole, introducing correlations between among ODEs through conditional probability modeling. During the training stage, the number and elements of the condition are randomly sampled to ensure the model's generalization. During the inference phase, conditions are added one by one.

$$\max_\theta \; \log \; P_\theta(\mathbf{y}|X) = \log \prod_{i=1}^{D} P_\theta(\mathbf{e}'_i|X, \mathbf{c}_i) = \sum_{i=1}^{D} \log \; P_\theta(\mathbf{e}'_i|X, \mathbf{c}_i) \tag{5}$$

Comparing three strategies, the universal strategy, without incorporating any prior knowledge, allows the model to autonomously learn and explore the physical constraints $R$ within KEP, resulting in pattern rigidity and lack of generalization ability. The other two focus on modeling each individual ODE in KEP. Independent strategy ignores the mutual constraints among ODEs, with each equation adopting an optimal solution. However, the predicted ODEs do not satisfy the constraint $R$ ($R \nvDash \mathbf{f}$). Conditional strategy effectively captures physical constraints $R$ by directly modeling the correlations among ODEs. Compared with the former two, it provides a generalizable framework for KEP. However the challenge of the generation order still remains.

## 4.2 GENERATION ORDER SEARCH

To solve the order challenge, we introduce a novel Monte Carlo Tree Search (MCTS) search module into KinFormer based on conditional strategy, as shown in Figure 3. Different from previous works (Holt et al., 2023; Shojaee et al., 2023; Li et al., 2024), the proposed search module does not

focus on the generation order of each token, but aims to coordinate with a conditional strategy to search for the generation order of individual ODEs. Therefore, nodes of Monte Carlo the represent independent equations ($\mathbf{e}'$), while edges reflect correlations between two equations. To complete an iteration, the MCTS search module includes four major stages: Selection, Expansion, Simulation, and Backpropagation (Silver et al., 2017).

**Selection**: When given the current activated node $\mathbf{e}'_i$, the next activated node should be selected from $\mathbf{e}'_i$'s child nodes. In this work, the probabilistic upper confidence bound heuristic (P-UCB) (Silver et al., 2018) is adopted to execute the selection operation. In details, the node with the maximum P-UCB score ($S$) is activated. The calculation of $S$ is in the Equation 6, where $\mathbf{e}'_{i+1} \in \text{child}(\mathbf{e}'_i)$ is the next activated node, one of the child nodes of $\mathbf{e}'_i$, $alpha$ is a constant, $P_\theta$ is the probability calculated by the neural model, $\mathbf{c}_{i+1} = [\mathbf{e}'_1, \cdots, \mathbf{e}'_i]$ is the condition. In MCTS, $V(\cdot)$ and $N(\cdot)$ are two important indicators to record the cumulative score and the number of traversals, respectively.

$$\mathbf{S}(\mathbf{e'_{i+1}}) = \frac{V(\mathbf{e}'_{i+1})}{N(\mathbf{e}'_{i+1})} + \alpha \cdot P_\theta(\mathbf{e}'_{i+1}|X, \mathbf{c}_{i+1}) \cdot \sqrt{\frac{\ln N(\mathbf{e}'_i)}{1 + N(\mathbf{e}'_{i+1})}}, \ \mathbf{e}'_{i+1} \in \text{child}(\mathbf{e}'_i) \quad (6)$$

**Expansion**: When the activated node $\mathbf{e}'_i$ is a leaf node, to facilitate tree growth, the expansion operation generates all child nodes (the next possible equations) of the current one. In this work, all untraversed ODEs $\mathbf{n}$ can be predicted by Equation 7, which is the expansion of $\mathbf{e}'_i$.

$$\mathbf{n} = \arg\max_{\mathbf{n}} P_\theta(\mathbf{n}|X, \mathbf{c}_i \cup \{\mathbf{e}'_i\}), \ \mathbf{n} \in \text{child}(\mathbf{e}'_i) = \mathbf{y}/(\mathbf{c}_i \cup \{\mathbf{e}'_i\}) \quad (7)$$

**Simulation**: Given an activated node $\mathbf{e}'_i$, the proposed module employs an action to random select a sequence of the rest ODEs and executes the action through the neural model step by step, resulting in a simulation result $\mathbf{y}'_i$.Then, the evaluation score $score$ can be computed by Equation 8, where $r2_m$ and $r2_M$ are two R2 metrics that will be defined in the next section, $\alpha + \beta = 1, \alpha \geq 0, \beta \geq 0$.

$$score(\mathbf{e}'_i) = \alpha \cdot r2_m(\mathbf{y}'_i, X) + \beta \cdot r2_M(\mathbf{y}'_i, X) \quad (8)$$

**Propagation**: After simulation, the activated node's score is recursively backpropagated to its parent node until reaching the root node ($N = N + 1$ and $V = V + score$).

We set iterations $n = 100$ for our MCTS framework. After removing error nodes, the leaf node with the maximum $N$ ($Q$ is used for equal $N$ values) is selected as the final ODEs. The detailed MCTS algorithms and the specific example are shown in the Appendix.

## 5 EXPERIMENTS

### 5.1 EXPERIMENTAL SETTINGS

For three training strategies, we construct different formats of training data, as shown in Figure 2. Because twenty types of reactions belong to four chemical mechanism categories mentioned in 3.2 and reactions under the different categories possess the distinct patterns, we have defined two distinct modes: **Intra-class** and **inter-class** generalization. **Intra-class**: randomly hold out one reaction type from bicatalytic mechanism, catalyst activation and catalyst deactivation (e.g. M5,M6 and M20). **inter-class**: hold out all three reaction types within catalyst activation (e.g. M6-M8). The detail of data sample construction is presented in Appendix C. We mainly compare KinFormer with three training strategies and two classical SR methods, namely SINDy (Brunton et al., 2016) and PySR (Cranmer, 2023). Although SINDy and PySR are usually plug-and-play, and do not possess pre-training and the concept of "generalization", we still conduct comparison experiments to prove the superiority of KinFormer. Following the previous work (d'Ascoli et al., 2023), we compute $\frac{\mathrm{d}}{\mathrm{d}t}\mathbf{x}(t)$ by the central finite difference algorithm as regression targets. Each test sample corresponds to one ODE system, so we apply SINDy or PySR to each sample and test each individual model. In SINDy, only the polynomial dimension ($d$) of hyper-parameters significantly influences the results, so we exhibit the performance of $n \in \{2, 3, 4, 5\}$. In PySR, we keep the default hyper-parameters setting. To further illustrate effectiveness MCTS, we also perform the equation-level beam search to replace

MCTS. However, as beam search is time-consuming, we set the iteration $n = 20$, sample 100 cases for each type of mechanism, and keep KinFormer under the same setting for a fair comparison. It is worth noting that the comparison between the conditional strategy and KinFormer can be regarded as the ablation study of the MCTS module.

## 5.2 EVALUATION METRICS

In this work, we adopt $R2$-based measurement and $RMSE$ as our evaluation metrics. $R2$-score (Cava et al., 2021) is a classical metric for regression tasks. $R2$-score is unbounded from below and a single outlier prediction causes severely biased score. To circumvent this, we set $0$ for any negative $R2$-score (Kamienny et al., 2022; d'Ascoli et al., 2023). Based on the definition of $R2$-score, we define four corresponding metrics to evaluate models. The first two are micro-R2 ($r2_m$) and macro-R2 ($r2_M$). $r2_m$ is the average $R2$-score on the test set. Due to the smaller scales of certain species (e.g. intermediate) compared with other species, $r2_m$ might overlook the significance of these species. Instead, $r2_M$ calculates the $R2$-score for each generated species and then take the average. The latter two are accuracy scores. We consider a sample with $R2$-score greater than $0.9$ to be a correct one. Values calculated by $R2_m$ and $R2_M$ are defined as micro-accuracy $Acc_m$ and macro-accuracy $Acc_M$, respectively. Furthermore, we also compare the accuracy of the equation form ($Acc_{form}$). Specifically, $Acc_{form}$ is calculated by exact match in form neglecting kinetic constants. $RMSE$ is calculated to measure the difference between the predicted and actual values. We report the median of RMSE to avoid the influence of outliers. $t$ is the average inference time.

## 5.3 MAIN RESULTS

Table 1 presents the main results of KinFormer and other baselines. SINDy and PySR does not perform as well as KinFormer. We speculate that one possible reason is that classical SR methods lack domain-specific knowledge (e.g. mass-action law) due to the absence of pre-training. Another reason probably is the imprecision of the approximated temporal derivatives. Comparing three training strategies, firstly, the universal strategy has performed optimally across all metrics in in-domain settings, demonstrates the power of Transformer on DSR. In both out-of-domain scenarios, however, it performs poorly, failing to predict the ODEs' form correctly. This indicates that the universal strategy lacks of generalization capability, merely memorizing formulas seen in training. Secondly, the independent strategy underperforms in all three scenarios, which is within expectations. When training a model to learn a specific equation of ODEs based on the entire concentration profiles, several irrelevant information is introduced and physical constraints from other equations are absent, which results in inaccurate predictions. Thirdly, the conditional strategy has made significant progress compared with the other strategies in two out-of-domain scenarios, even $Acc_{form}$ exceeding 70%. The improvement demonstrates its generalization capability and proving our assumption that the extra conditional information enables the model to successfully learn the physical constraints among equations. Last but not least, the model trained under the universal strategy have the fastest inference speed, but the difference among the three is not significant.

KinFormer incorporates the conditional strategy and the MCTS module, which has achieved the superior results. Specifically, compared with the universal strategy, in the intra-class scenario, $Acc_m$, $Acc_M$, $R2_m$ and $R2_M$ increase by 47.17%, 21.31%, 0.563 and 0.445, respectively; in the inter-class scenario, $Acc_m$, $Acc_M$, $R2_m$ and $R2_M$ increase by 30.42%, 18.52%, 0.399 and 0.410, respectively. Overall, the experimental results demonstrate that KinFormer ($n = 100$) significantly enhances generalization capability by implicitly modeling physical constraints through the conditional strategy and optimizing the sequential order of formulas generation via MCTS post-processing. Consequently, KinFormer is capable of predicting successfully for reaction types it has not previously encountered. To further illustrate the selection of MCTS instead of other search methods, we exhibit the performance of equation-level beam search. Under the same setting ($n = 20$), the inference time for Beam Search[1] is almost three times that of MCTS. Considering performance, MCTS algorithm consistently outperforms Beam Search in terms of both accuracy and efficiency. These results indicate that MCTS leverages the physical correlations learnt by conditional strategy to search the better generation sequence. In addition, disregarding the differences in data volumn, MCTS with 100

---

[1] $n = 20$ for Beam Search means we retain at most the top 20 equations after each equation generating step (e.g. generate $[dx0, dx3]$ from $[dx0]$ or generate $[dx0, dx3, dx4]$ from $[dx0, dx3]$ ) and end up with 20 ODEs.

Table 1: Evaluation Results of KinFormer and other baselines. *ID*, *OOD(Intra)* and *OOD(Intra)* represent in-domain, intra-class, and inter-class evaluations, respectively. The **bolding methods** are proposed in this work. $*$ means a summary result of individual models for each test sample is reported. Underline "$\underline{n=20}$" means that test set contains 100 samples for each type of mechanism.

| | Methods | $Acc_m$ | $Acc_M$ | $Acc_{form}$ | $R2_m$ | $R2_M$ | $RMSE$ | $t(s)$ |
|---|---|---|---|---|---|---|---|---|
| | SINDy$*$ | | | | | | | |
| | *d=2* | 1.59 | 0.72 | 0 | 0.020 | 0.042 | 174.9 | 1 |
| | *d=3* | 0.09 | 0.04 | 0 | 0.002 | 0.032 | 5.252 | 1 |
| | *d=4* | 0 | 0 | 0 | 0.001 | 0.028 | 1.175 | 1 |
| | *d=5* | 0 | 0 | 0 | 0.003 | 0.042 | 1.278 | 1 |
| | PySR$*$ | 24.32 | 7.85 | - | 0.315 | 0.434 | 0.063 | 36 |
| *ID* | Univeral | **71.49** | **38.02** | **67.80** | **0.754** | **0.689** | 0.008 | 6 |
| | Independent | 4.14 | 1.86 | 0 | 0.059 | 0.133 | 0.248 | 10 |
| | **Conditional** | 36.08 | 14.78 | 57.15 | 0.471 | 0.447 | 0.018 | 15 |
| | BeamSearch | | | | | | | |
| | $\underline{n=20}$ | 35.48 | 17.70 | 63.60 | 0.463 | 0.455 | 0.010 | 669 |
| | **KinFormer** | | | | | | | |
| | $\underline{\textbf{n=20}}$ | 63.45 | 28.23 | 64.99 | 0.726 | 0.638 | **0.007** | 222 |
| | $\textbf{n=100}$ | 59.76 | 27.56 | 64.00 | 0.698 | 0.629 | 0.010 | 371 |
| | Universal | 10.40 | 1.87 | 0 | 0.158 | 0.183 | 0.095 | 6 |
| | Independent | 0.47 | 0.07 | 0 | 0.009 | 0.077 | 0.304 | 10 |
| | **Conditional** | 31.73 | 9.67 | 70.28 | 0.455 | 0.425 | 0.023 | 15 |
| *OOD* | BeamSearch | | | | | | | |
| *(Intra)* | $\underline{n=20}$ | 31.67 | 12.67 | 73.19 | 0.498 | 0.469 | 0.017 | 669 |
| | **KinFormer** | | | | | | | |
| | $\underline{\textbf{n=20}}$ | 50.83 | 18.75 | 79.42 | 0.662 | 0.583 | 0.014 | 222 |
| | $\textbf{n=100}$ | **57.57** | **23.18** | **80.47** | **0.721** | **0.628** | **0.013** | 371 |
| | Universal | 10.73 | 0.60 | 0 | 0.182 | 0.205 | 0.067 | 6 |
| | Independent | 0.47 | 0 | 0 | 0.012 | 0.075 | 0.322 | 10 |
| | **Conditional** | 23.40 | 4.40 | 74.30 | 0.325 | 0.397 | 0.044 | 15 |
| *OOD* | BeamSearch | | | | | | | |
| *(Inter)* | $\underline{n=20}$ | 23.00 | 6.67 | 76.05 | 0.325 | 0.404 | 0.021 | 669 |
| | **KinFormer** | | | | | | | |
| | $\underline{\textbf{n=20}}$ | 40.47 | 13.71 | 80.61 | 0.515 | 0.534 | 0.015 | 222 |
| | $\textbf{n=100}$ | **41.15** | **19.12** | **81.41** | **0.581** | **0.615** | **0.012** | 371 |

iterations shows only a marginal improvement over MCTS with 20 iterations, which supports that KinFormer can coverage within a small number of iterations, demonstrating high search efficiency.

## 5.4 GENERALIZATION ANALYSIS

We plot distribution diagrams of the $R2$-score under both intra-class and inter-class conditions, shown in Figure 4. The distribution that is more skewed towards the right indicates a superior fitting performance, implying a better generalization. Since $r2_m$ is to evaluate the overall performance and $r2_M$ is to assess the individual equation fitting performance, The analysis of the diagram can be categorized into four scenarios: (1) large $r2_m$ and $r2_M$ mean great fitting; (2) small $r2_m$ and $r2_M$ mean poor fitting; (3) large $r2_m$ and small $r2_M$ mean overall good fitting but neglect of a few species due to tiny scaling; (4) small $r2_m$ and large $r2_M$ means only a few species are predicted accurately.

For the intra-class scenario, it is evident from Figure 4a that the universal exhibits mediocre performance on M5. Distributions for both M6 and M20 are skewed notably towards the left, suggesting a

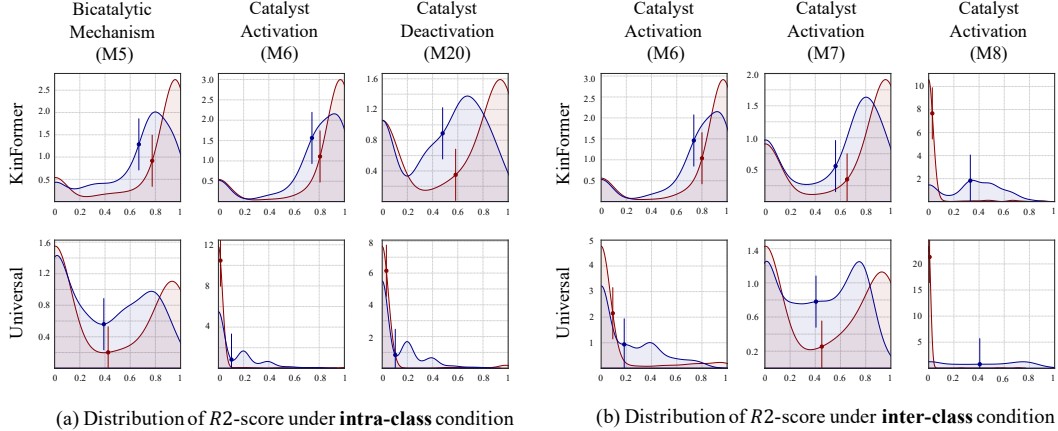

Figure 4: Distribution diagrams of $R2$-score under out-of-domain conditions. **Red** curve represents $r2_m$ and **Blue** curve represents $r2_M$. The vertical line represents mean value of $R2$-score.

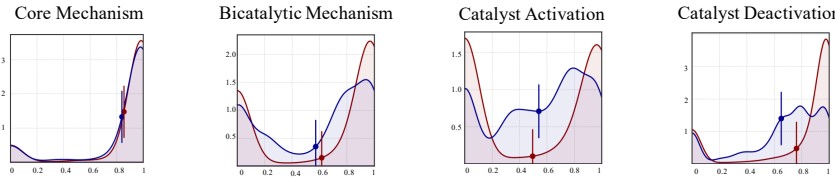

Figure 5: KinFormer distribution diagrams of $R2$-score under in domain conditions.

inability to generalize to these two mechanisms. However, KinFormer successfully shifts this distribution towards the right, demonstrating its capability to effectively generalize under the intra-class scenario. The inter-class scenario is harder setting because the whole Catalyst Activation haven't been seen during training. As evidenced in Figure 4b, KinFormer achieves superior performance on both M6 and M7. Specifically, KinFormer improve the $r2_M$ distribution to the right obviously, which means KinFormer not only enhances the overall ODEs fitting performance, but also promote the species profiles fitting with small numerical scale. KinFormer and the universal strategy both fail on M8. We speculate that The dimension of most mechanisms are either 4 or 5, while the dimension for M8 is 6, thereby posing a significant challenge in prediction. Meanwhile, KinFormer on M20, the universal strategy on M5 and M7 exhibit clear mismatches of the two $R2$-score, suggesting that the model still tends to overlook species at smaller scales.

## 5.5 MECHANISM CATEGORY ANALYSIS

We specifically analyze the performance of KinFormer based on the four reaction mechanism categories under in domain setting. From Figure 5, firstly, it is obvious that KinFormer achieves the superior performance in **Core Mechanism**, because the first mechanism is relatively simple, whose dimension is only four and form is basic. Secondly, for **Bicatalytic Mechanism**, dimensions increase to five/six, raising the difficulty of the task. However, KinFormer still provides the moderate results, where $r2_m$ exceeds 0.6, which proves the effectiveness of KinFormer. Thirdly, **Catalyst Activation Mechanism** is the most difficult to predict with the lowest $R2$-score. Meanwhile, the two peaks of $r2_m$ of the category are close to 0 and 1, but its $r2_M$ is in the middle, indicating that the scenarios (3) and (4) mentioned in the previous section are both present. Fourthly, for **Catalyst Deactivation Mechanism**, which is also a challenging task, the distribution of $R2$-score is obviously shifted to the left and $r2_m$ is almost 0.8. In addition, Kinformer also neglects small-scale species in this category.

## 5.6 GENERATION ORDER ANALYSIS

The top three generated sequences of MCTS and frequency count for each reaction category under in-domain scenario are shown in Appendix. It is worth noting that almost all reactions of catalyst

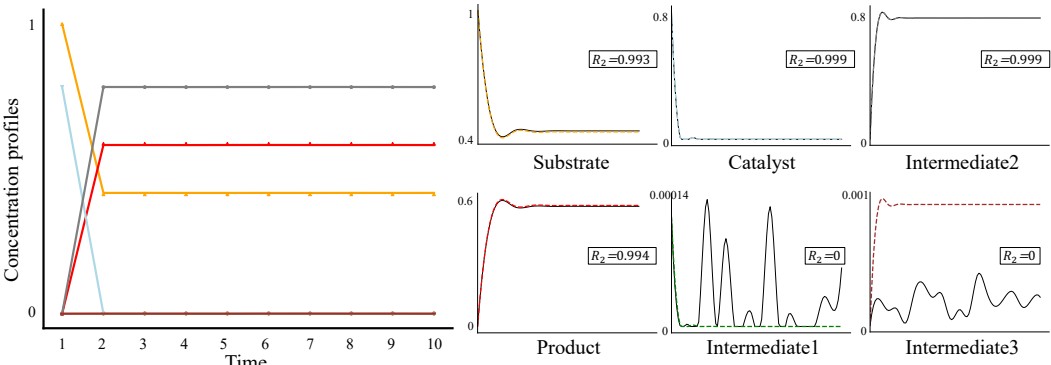

Figure 6: Case study on M20 to illustrate E2 errors. LHS part is complete concentration profiles and predicted profiles of six involved species. The individual subfigures are presented in RHS part. Different color represents different species. Black represents ground-truth profiles.

deactivation prefer a specific generation order: $[O, P, catS, S, cat]$, where $O$ represents another intermediate distinct from $catS$. In fact, except for $O$ and $P$, the ODE coefficients of other neighboring species are correlated (e.g. in Figure 1, the $k_1[S][cat]$ term appears simultaneously in the equations of $d[S]$ and $d[cat]$). Therefore, the relative sequence are more important and we infer that KinFormer tends to select a relative generation order that makes terms and independent coefficients between two consecutive ODE equations similar.

## 5.7 ERROR ANALYSIS AND CASE STUDY

The prediction errors of KinFormer mainly stem from the predictions of M3, M8, M10, M15, and M20. From a dimensional perspective, the errors on M8, M10, M15, and M20 originate from their highest dimensionality ($D = 6$), indicating that KinFormer's ability for cross-dimensional prediction needs improvement. When classifying error types, errors on M3 and M8 belong to the category of unpredictable errors (E1), while errors on M10 and M15 belong to the category of overlooking small-scale species errors (E2). The errors on M20 exhibit characteristics of both categories. We further perform a case study to illustrate E2. In Figure 6, the case from M20 has relatively high $r2_m$ but low $r2_M$. Since the reaction reaches a steady state within the first 10 timesteps, we only took the first 10 timesteps. This is a typical E2 error. From Figure 6, Kinformer cannot effectively predict Intermediate1 and Intermediate3. Digging into the reasons, the ground truth of these species' concentration profiles is severely disrupted by Gaussian noise. KinFormer is unable to reconstruct the expressions from noisy data, especially when the noise magnitude is similar to the data scale. This indicates that E2 errors stem from species severely affected by noise.

## 6 LIMITATIONS

Limitations include: (1) The performance both in- and out-of-domain needs further improvement. (2) The cross-dimensional transfer capability is weak. (3) MCTS faces the issue of high time complexity when dealing with a large number of species.

## 7 CONCLUSIONS

Kinetic equation prediction is an essential task for understanding the chemical reaction mechanisms. Traditional DSR models fail to provide generalizable performance under chemical domain. In this paper, we propose KinFormer to solve the generalization problem in KEP, which promotes the development of chemical kinetics analysis. Specifically, KinFormer takes advantage of the conditional training strategy to model DSR under physical constraints. Meanwhile, MCTS is applied as a post-processing step to select the optimal generation order. Experiments on a synthetic catalytic organic reaction dataset demonstrate KinFormer's generalization capability. In conclusion, KinFormer is a generalizable DSR framework for KEP, which can inspire future research in other scientific domains.

## 8 ACKNOWLEDGE

We thank SJTU AI for Science platform for the computing support. This work was supported by the Shanghai Municipal Science and Technology Major Project (2021SHZDZX0102), and the Fundamental Research Funds for the Central Universities.

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

APPENDIX

## A  KINETIC MECHANISM OF TWENTY CATALYTIC ORGANIC REACTIONS

Mechanism diagram describes the species transformation relations under a specific chemical reaction. Taking Figure 1 as an example, the reaction is $S \xrightarrow{cat} P$ and includes several elementary reactions represented by arrows with different color , such as $S + cat \leftrightarrow catS$. The elementary reaction is reversible, with both the outer ring and inner ring representing this elementary reaction. The outer ring represents the forward process, which is the primary and represents the $cat$ combining with the $S$. The inner ring represents the reverse process, which is secondary and represents the decomposition of $catS$. The mass-action law is a fundamental concept in chemical kinetics, describing the rate of an elementary reaction is proportional to the product of the concentrations of involved species. For example, equation $\frac{dS}{dt}$ is decided by two parts: 1) combination of $S$ and $cat$; 2) decompositon of $catS$; which are shown by red arrow and purple arrow, respectively. Therefore, $\frac{dS}{dt} = k_{-1}[catS] - k_1[S][cat]$. Each type of reaction mechanism is defined by the reaction diagram. The ordinary differential equations (ODEs) correspond to each diagram, where the kinetic constants control the rate of change of each species in the reaction.

Figure 7 illustrates the core mechanism, bicatalytic mechanisms, and catalyst activation mechanisms. The core mechanism is the basis of catalytic organic reaction. It demonstrates the process that the substrate combines with the catalyst to form an intermediate, which subsequently decomposes into the product. Bicatalytic reactions include additional steps of the combination of two catalyst molecules to facilitate the reaction. In catalyst activation reactions, it is necessary to activate the catalyst. Figure 8 and Figure 9 demonstrate catalyst deactivation mechanisms. In contrast to catalyst activation, catalyst deactivation reactions inevitably lead to the generation of deactivated catalysts.

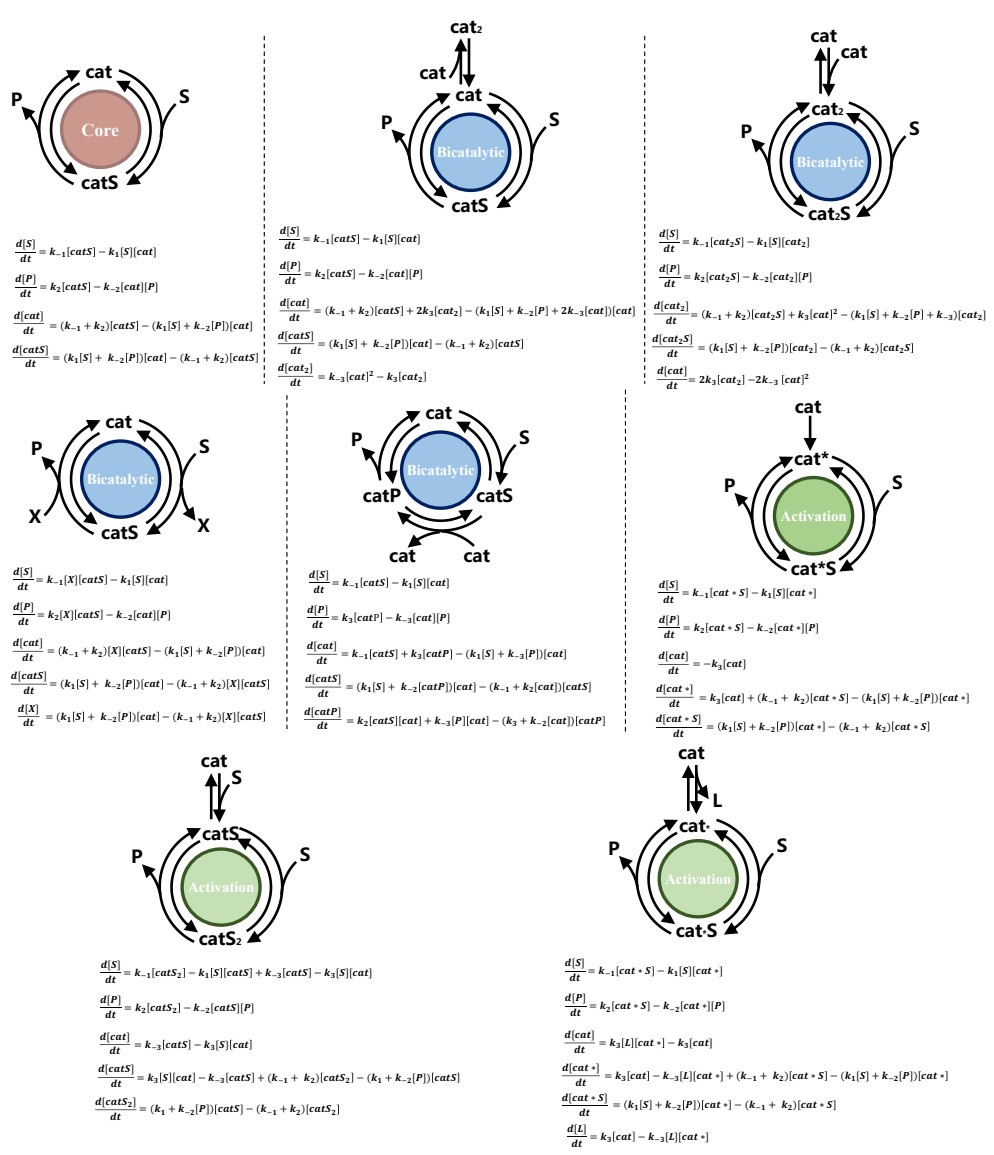

Figure 7: Core mechanism, bicatalytic mechanism and catalyst activation diagrams and corresponding ODEs (M1-M8)

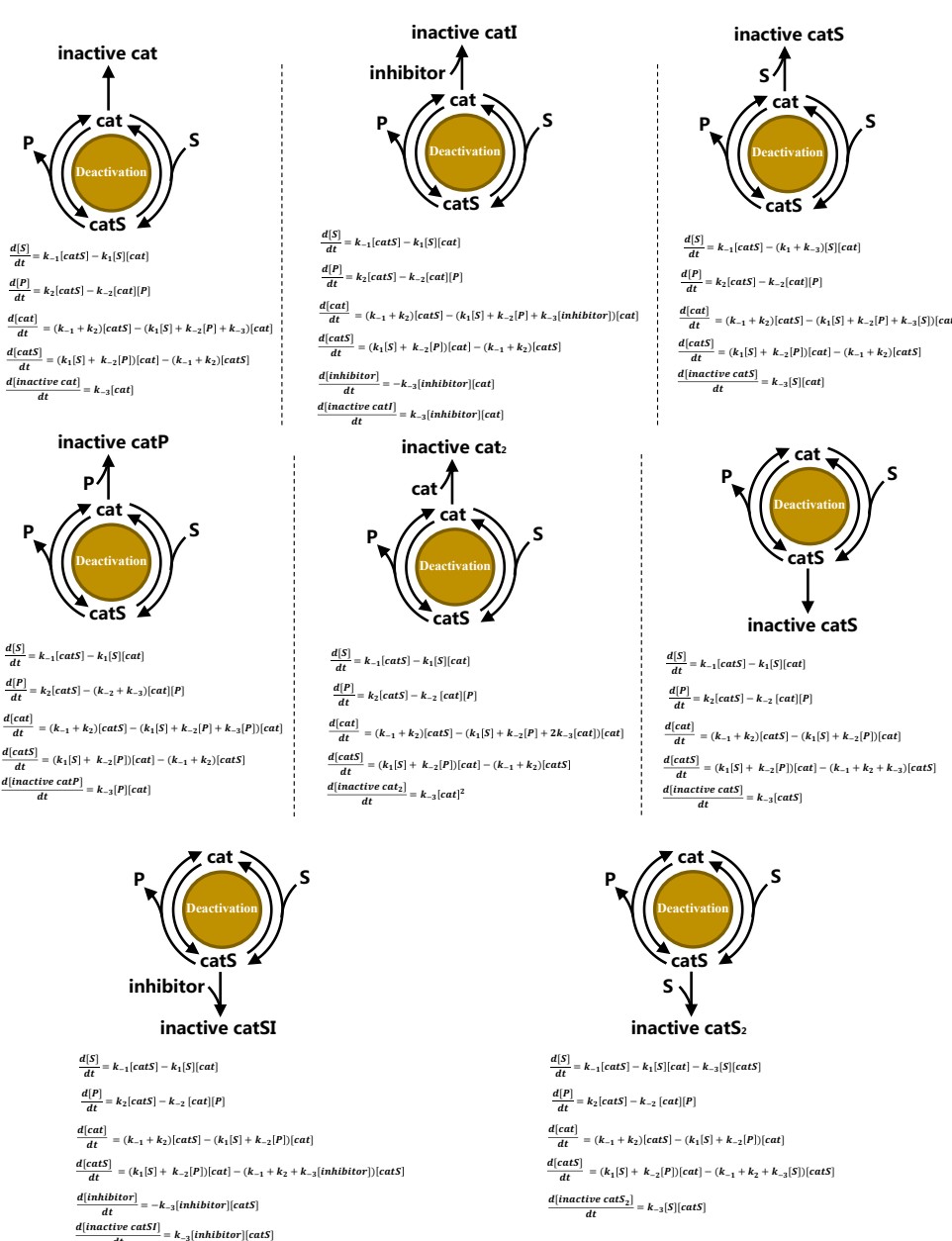

Figure 8: Catalyst deactivation diagrams and corresponding ODEs (M9-M16)

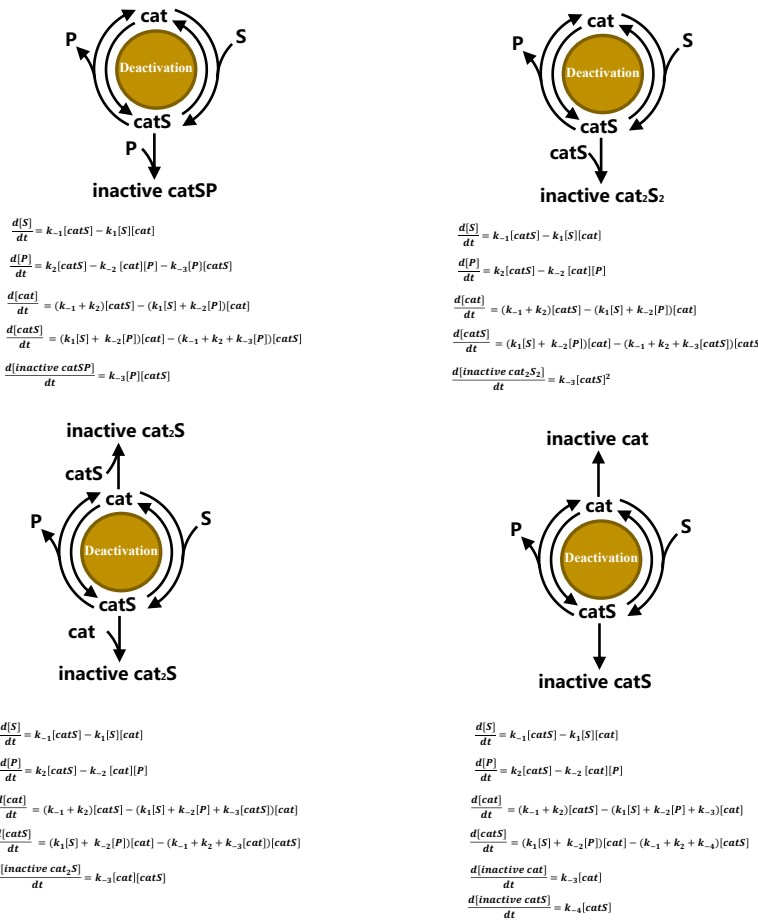

Figure 9: Catalytic deactivation diagrams and corresponding ODEs (M17-M20)

## B    GENERATION ORDER SEARCH WITH MCTS

---

**Algorithm 1:** Generation Order Search

---

**Input:** Kinetic data $X$, conditonal model $P_\theta$, hyper-parameters $\alpha$, $\beta$, number of iterations $n$, constant $c$

**Output:** ODEs $y$

1  # Initialize the tree and execute the first expansion operation.

2  **initialize** the tree $tr$ with the root $r$ ;

3  **initialize** $N(r) = 0, V(r) = 0$ ;

4  $nodes = r.\text{expand}(P_\theta, X)$ ;

5  $tr.\text{update}(nodes, r)$ ;

6  # Execute MCTS. **for** $i \leftarrow 1$ **to** $n$ **do**

7  |    $tr = \text{MCTS}(tr, r, X, P_\theta, \alpha, \beta, c)$

8  **end**

9  # Generate ODEs from the Monte Carlo tree.

10  $y = (\arg\max_{n \in tr.\text{leafNodes}()} N(n)).\text{getEqs}()$;

11  **return** $y$

---

The execution of generation order search is shown in Algorithm 1. Given the kinetic data $X$ and conditional model $P_\theta$, the first step is to initialize the tree and execute the first expand operation. Afterwards, MCTS is executed. Algorithm 2 shows how to adopt MCTS to search the best generation

---

**Algorithm 2:** MCTS

---

**Input:** Monte Carlo tree $tr$, activated node $n_{act}$, Kinetic data $X$, conditonal model $P_\theta$; hyper-parameters $\alpha$, $\beta$, constant $c$

**Output:** Updated tree $tr$

1 **if** $n_{act} == tr.\text{getRoot}()$ **then**
2     $n_{act} = tr.\text{select}(n_{act}, c)$ ;
3     $tr = \text{MCTS}(tr, n_{act}, X, P_\theta, \alpha, \beta, c)$
4 **else**
5     **if** $tr.\text{getChidren}(n_{act}) == []$ **then**
6        **if** $N(n_{act}) == 0$ **then**
7           $score = tr.\text{simulate}(n_{act}, X, P_\theta, \alpha, \beta)$ ;
8           $tr.\text{backpropagate}(n_{act}, score)$ ;
9           $tr.\text{update}(N, V)$
10        **else**
11           $nodes = n_{act}.\text{expand}(P_\theta, X)$ ;
12           $tr.\text{update}(nodes, n_{act})$ ;
13           $n_{act} = tr.\text{select}(n_{act}, c)$;
14           $score = tr.\text{simulate}(n_{act}, X, P_\theta, \alpha, \beta)$ ;
15           $tr.\text{backpropagate}(n_{act}, score)$ ;
16           $tr.\text{update}(N, V)$
17        **end**
18     **else**
19        $n_{act} = tr.\text{select}(n_{act}, c)$;
20        $tr = \text{MCTS}(tr, n_{act}, X, P_\theta, \alpha, \beta, c)$
21     **end**
22 **end**
23 **return** $tr$

---

order, which is a recursive algorithm. When the activated node is not the root, it is necessary to determine whether it is a leaf node. If the activated node is not a leaf node, continue to select the next node and perform recursion; otherwise, continue to judge whether the current node has been traversed. If the answer is "no", execute simulate operation and then backpropagate to the root node; otherwise, execute expand operation and then select the next node. After $n$ iterations, select the sequence with the largest $N$ as the final prediction.

Specifically, our MCTS algorithm is at the equation level rather than the token level. Taking ODEs in Figure 1 as an example, there are four eqaution components, denoted as $dx0$, $dx1$, $dx3$ and $dx4$, respecitvely. During initialization, we first allow the model to independently generate these four equation components as initial nodes. During selection, we use P-UCB (equation 6) to select the specific eqaution component (e.g. $dx1$ node). Because this node has not been traversed yet, we will proceed directly with the simulation. During simulation, we use the $dx1$ equation as the condition to generate the next equation (e.g. $dx0$). Then we combine $dx1$ and $dx0$ as condition to generate the next, and so on. Simulation will end until we get the full ODEs. After that, by solving ODEs, we can obtain numerical solutions for each reaction species. These numerical solutions can then be compared with the input data (e.g. temporal concentration profiles) to calculate the R2 score. During backpropagation, we compute the reward according to R2 score and then record it as the node value. When the $dx1$ node has been traversed, we will do expansion. During expansion, we use the $dx1$ equation as condition to generate all possible next equations (e.g. $dx0$, $dx2$ or $dx3$ ), respectively. The above processes are executed iterately. This is an easy example of our MCTS framework.

## C  DATASET CONSTRUCTION

Considering the requirement for data authenticity and large data volume, we construct a simulated dataset based on the methods described in Nature work (Burés & Larrosa, 2023). First, based on the categorized mechanism types (20 in total), we identify the corresponding ODEs. To construct valid ODEs, We randomly pick all kinetic constants in the range of $10^5$-$10^{-5}$ A.U. and rounded

to three significant figures. The chosen kinetic constants must fulfill the conditions defined for their corresponding mechanism and allow for reversible and irreversible reactions with maximum yields between 20% and 90% to match actual experimental records. The initial concentration of reactant is kept as 1 and the product is 0. The catalyst's initial state is sampled between 0.01 and 0.99. Then, we use *solve_ivp* from the scipy Python package based on a LSODA solver to obtain reaction concentration profiles as input data. It is worth noting that in order to construct a computable ODEs, we utilize Networkx to construct a tree structure diagram of the formula, which is used for computation with scipy. The complete symbolic mathematical expressions of ODEs is transformed into prefix notation sequence and are kept as training label. We have eliminated all generated equations that are unsolvable and equations where the rate of change in intermediate products is less than 0.2. Eventually, for each type of catalytic reaction, we generate 5,000 reaction samples for training and 500 reaction samples for testing, which results in 100,000 training data and 10,000 test data. Based on our data scale, we introduce the Gaussian noise to each training sample with a standard deviation of $1e$-4 to improve model's robustness.

For different traning strategies, we choose different data format. Specifically, for universal transformer, we keep time series $\{(t_i, \mathbf{X}_i)\}_{i \in [1,2,...,T]}$ as input to encoder and the model is taught to predict the complete ODEs of a reaction sample. For independent transformer, we split ODEs into several independent equations and append an index prompt (e.g. $dx0$ for the first ODE of Substrate) after time series as encoder input. The independent ODE corresponding to the index prompt is utilized to train the model. For conditional transformer, we keep one of ODEs from each reaction sample as training label and randomly choose several left ODEs as condition. Combining corresponding index prompts, the model is trained to predict one ODE according other ODEs condition and learn the implicit constraints between ODEs.

**Why Shuffling order**: Firstly, KinFormer aims to capture physically-motivated correlations established by mass-action law, instead of searching for the optimal, deterministic, physically-meaningful generation order. Secondly, Shuffling the order is helpful for the model to grasp physically-motivated correlations instead of spurious correlations. Actually, the attention mechanism of Transformer Decoder can capture these inner correlations implicitly, but tends to be negatively affected by spurious correlations. For example, in Figure 1, the $k_1[S][cat]$ term appears simultaneously in the equations of $d[S]$ and $d[cat]$, indicating the physically-meaningful correlation from the mass-action law; The fixed generation order of $(d[S], d[P], d[cat], d[catS])$ shows the spurious correlation, which is not necessary. If the order is not shuffled, the Transformer Decoder is likely to capture such a fixed generation sequence rather than physically-motivated correlations, thereby making it difficult to generalize to unseen reaction types.

## D  TRAINING DETAILS

Hyper-parameters and training settings of three strategies are shown in Table D. It can be seen that almost all settings are shared by three strategies, except **Equations/Epoch** and **Batch size**. **Equations/Epoch** means the number of the equation sampled in each epoch. For the independent strategy and KinFormer (and the conditional strategy), **Equations/Epoch = Batch size × Steps Per Epoch** holds because these models generate only one ODE for each step (sampling 4 equations for each set of ODEs). However, the universal strategy generates the whole set of ODEs so the total number of equation is different. Considering **Batch size**, due to GPU memory constraints, KinFormer adopts a smaller batch size. In principle, the total numbers of equations for each epoch are similar, although the universal strategy introduces more equations.

## E  DEFINITION OF EVALUATION METRICS

We define $r2_m$ to evaluate the overall performance of one test sample, that is, to compare the generated curves of all species changes as a whole with the original profiles:

$$r2_m = 1 - \frac{\sum_{k=1}^{D} \sum_{i=1}^{T} (x_i^k - \hat{x}_i)^2}{\sum_{k=1}^{D} \sum_{i=1}^{T} (x_i^k - \bar{x}_i)^2} \tag{9}$$

Table 2: Training settings.

| Settings | Universal | Independent | KinFormer |
|---|---|---|---|
| Embedding dim | 64 | 64 | 64 |
| Encoder dim | 256 | 256 | 256 |
| Decoder dim | 256 | 256 | 256 |
| Activation | silu | silu | silu |
| Loss function | CE | CE | CE |
| Dropout | 0 | 0 | 0 |
| Optimizer | Adam | Adam | Adam |
| Schedule | Cosine | Cosine | Cosine |
| Learning rate | $2e^{-4}$ | $2e^{-4}$ | $2e^{-4}$ |
| Warmup | 10000 | 10000 | 10000 |
| Clip | 1.0 | 1.0 | 1.0 |
| Weight decay | 0 | 0 | 0 |
| Dropout | 0 | 0 | 0 |
| Epochs | 100 | 100 | 100 |
| Equations/Epoch | 128750 | 100000 | 100000 |
| Batch size | 50 | 200 | 100 |
| $c$ | − | − | 1.0 |
| $\alpha$ | − | − | 0.5 |
| $\beta$ | − | − | 0.5 |
| Device | NVIDIA 3090 24G | NVIDIA 3090 24G | NVIDIA 3090 24G |

Due to the minimal changes of certain species during the reaction (such as intermediate), the $r2_m$ calculation might overlook the significance of these species. Therefore, we define $r2_M$ to assess the individual fitting performance, that is, to calculate the $R2$-score for each generated species profile independently, and then take the average:

$$r2_M = \frac{1}{D} \sum_{k=1}^{D} \left[ 1 - \frac{\sum_{i=1}^{T}(x_i^k - \hat{x}_i^k)^2}{\sum_{i=1}^{T}(x_i^k - \bar{x}_i^k)^2} \right] \tag{10}$$

The reported $R2$-score is calculated by averaging all test samples:

$$R2_m = \frac{1}{N_{test}} \sum_{j=1}^{N_{test}} r2_m^j; \quad R2_M = \frac{1}{N_{test}} \sum_{j=1}^{N_{test}} r2_M^j \tag{11}$$

We consider a test sample with $R2$-score greater than the threshold of $0.9$ to be correct due to great fitting performance:

$$Acc_m = \frac{1}{N_{test}} \sum_{j=1}^{N_{test}} \mathbb{1}\left[ r2_m^j > 0.9 \right]; \quad Acc_M = \frac{1}{N_{test}} \sum_{j=1}^{N_{test}} \mathbb{1}\left[ r2_M^j > 0.9 \right] \tag{12}$$

## F    EXTRA RESULTS

Table 3 shows the results of KinFormer on all twenty types of mechanisms and Figure 9 shows their distributions of $R2$-scores. KinFormer perform well on M1, M4, and M12.

It can be observed that in most mechanisms, the KinFormer performs quite well. The $r2_m$ consistently exceeds 0.6, and even surpasses 0.8 in some mechanisms. However, it is noteworthy that the performance of the mechanisms in M3 and M8 is not satisfactory. We have analyzed the following

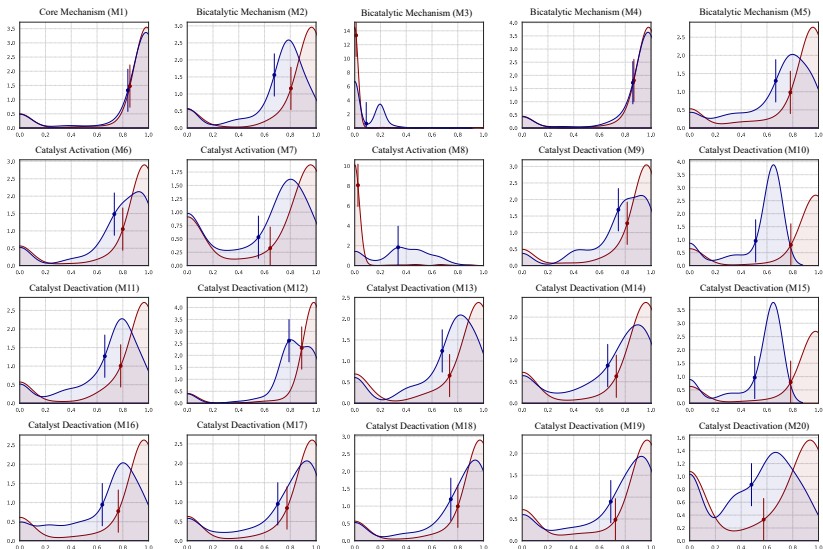

Figure 10: KinFormer distribution diagrams of R2-score for all twenty mechanism types under in domain conditions. **Red** curve represents $r2_m$ and **Blue** curve represents $r2_M$. The vertical line represents mean value of $R2$-score.

Table 3: Results of KinFormer on twenty types of mechanisms.

| Mechanism | $Acc_m$ | $Acc_M$ | $Acc_{form}$ | $R2_m$ | $R2_M$ |
|---|---|---|---|---|---|
| M1 | 82.8 | **76.8** | 88.8 | 0.855 | 0.841 |
| M2 | 68.0 | 17.0 | 52.2 | 0.805 | 0.679 |
| M3 | 0.6 | 0 | 49.2 | 0.012 | 0.088 |
| M4 | 82.0 | 76.4 | **91.0** | 0.870 | **0.856** |
| M5 | 61.8 | 22.8 | 77.6 | 0.779 | 0.672 |
| M6 | 67.6 | 39.6 | 88.2 | 0.799 | 0.734 |
| M7 | 53.0 | 17.2 | 68.4 | 0.648 | 0.553 |
| M8 | 0.2 | 0 | 81.0 | 0.028 | 0.337 |
| M9 | 66.6 | 36.0 | 74.0 | 0.812 | 0.745 |
| M10 | 71.4 | 0 | 31.0 | 0.786 | 0.512 |
| M11 | 63.6 | 17.6 | 30.6 | 0.785 | 0.663 |
| M12 | **85.2** | 39.6 | 58.0 | **0.887** | 0.791 |
| M13 | 61.6 | 24.2 | 63.8 | 0.739 | 0.678 |
| M14 | 59.0 | 30.6 | 44.8 | 0.733 | 0.665 |
| M15 | 70.0 | 0 | 56.4 | 0.783 | 0.507 |
| M16 | 65.2 | 18.8 | 60.0 | 0.768 | 0.643 |
| M17 | 64.8 | 42.4 | 65.0 | 0.771 | 0.702 |
| M18 | 69.6 | 45.6 | 53.0 | 0.801 | 0.742 |
| M19 | 62.6 | 39.4 | 70.4 | 0.726 | 0.688 |
| M20 | 39.6 | 7.2 | 76.6 | 0.573 | 0.481 |

reasons. In the M3 mechanism, the formula includes a squared term, which is very uncommon in other types. This contributes to the difficulty in learning. In the M8 mechanism, the dimension of ODEs increased to 6, which is uncommon in other types. This led to unsuccessful learning. In addition, $c$, $\alpha$ and $\beta$ are hyper-parameters only for MCTS.

Table 4: : R2 and RMSE quantile distribution. We keep the original R2 value, instead of replacing the negative value of R2 with zero in the paper. KinFormer outperforms than other baselines under two out-of-domain scenarios. Lower means lower quartile and Upper means upper quartile.

| Methods | | $R2$ | | | $RMSE$ | | |
|---|---|---|---|---|---|---|---|
| | | Lower | Median | Upper | Lower | Median | Upper |
| *ID* | Universal | **0.879** | **0.983** | **0.996** | **0.004** | **0.008** | **0.018** |
| | Independent | -72.633 | -22.456 | -4.596 | 0.137 | 0.247 | 0.412 |
| | **Conditional** | 0.402 | 0.915 | 0.986 | 0.007 | 0.018 | 0.041 |
| | **KinFormer** | 0.276 | 0.951 | 0.991 | 0.005 | 0.011 | 0.022 |
| *OOD (Intra)* | Universal | -16.977 | -1.271 | 0.803 | 0.028 | 0.095 | 0.239 |
| | Independent | -3942.36 | -38.409 | -8.605 | 0.193 | 0.304 | 2.431 |
| | **Conditional** | 0.343 | 0.881 | 0.973 | 0.011 | 0.023 | 0.045 |
| | **KinFormer** | **0.611** | **0.934** | **0.984** | **0.007** | **0.013** | **0.025** |
| *OOD (Inter)* | Univeral | -5.036 | -0.112 | 0.806 | 0.026 | 0.067 | 0.157 |
| | Independent | -7584.96 | -33.462 | -10.596 | 0.222 | 0.322 | 3.255 |
| | **Conditional** | -11.744 | 0.452 | 0.942 | 0.016 | 0.044 | 0.188 |
| | **KinFormer** | **-0.907** | **0.689** | **0.977** | **0.007** | **0.012** | **0.024** |

We also provide the extra results of $R2$ and $RMSE$ shown in Table 4. We report the lower quartile, median and upper quartile of metrics. We keep the original R2 value, instead of replacing the negative value of R2 with zero. The results are consistent with our conclusion in the paper.

The main packages of Python we used include: (1) Scipy 1.10.1 for ODE solving; (2) Networkx 2.8.2 for Monte Carlo tree construction; (3) Pytorch 2.0.1 for Transformer construction and training; (4) Scikit-learn 1.0.2 for model wrapping.

## G TOP 3 ORDERS FROM MCTS

Table 5 shows the top 3 orders from MCTS. It can be seen that bicatalytic reactions and catalyst activation do not exhibit a specific order. However, almost all reactions with catalyst deactivation share the same order: $[O, P, catS, S, cat]$, where $O$ represents intermediates distinct from $catS$.

Table 5: Top 3 generation orders statistics from MCTS. The number after generation order is corresponding frequency count.

| Mechanism | Top 1 | Top 2 | Top 3 |
|---|---|---|---|
| M1 | $[P, catS, S, cat]$:57 | $[S, cat, P, catS]$:54 | $[P, S, cat, catS]$:51 |
| M2 | $[O, P, catS, S, cat]$:21 | $[cat, O, P, S, catS]$:21 | $[catS, P, S, O, cat]$:19 |
| M3 | $[P, S, O, catS, cat]$:19 | $[cat, S, O, S, catS]$:18 | $[P, O, S, cat, catS]$:18 |
| M4 | $[P, O, S, cat, catS]$:30 | $[P, cat, S, catS, O]$:26 | $[S, O, P, cat, catS]$:23 |
| M5 | $[catS, S, O, P, cat]$:37 | $[O, P, catS, S, cat]$:29 | $[catS, S, cat, P, O]$:27 |
| M6 | $[cat, O, S, P, catS]$:21 | $[cat, O, P, S, catS]$:19 | $[cat, S, catS, P, O]$:19 |
| M7 | $[O, P, cat, S, catS]$:25 | $[catS, cat, S, P, O]$:20 | $[S, cat, catS, P, O]$:17 |
| M8 | $[cat, P, O, S, catS, O]$:29 | $[P, cat, O, S, catS, O]$:25 | $[P, cat, O, S, O, catS]$:25 |
| M9 | $[O, P, catS, S, cat]$:29 | $[O, S, catS, P, cat]$:27 | $[cat, O, S, P, catS]$:21 |
| M10 | $[O, O, P, catS, S, cat]$:14 | $[O, O, P, S, cat, catS]$:10 | $[O, O, P, catS, S, cat]$:9 |
| M11 | $[O, P, catS, S, cat]$:31 | $[O, P, cat, S, catS]$:24 | $[S, O, catS, P, cat]$:16 |
| M12 | $[O, P, catS, S, cat]$:49 | $[O, S, catS, P, cat]$:38 | $[O, P, cat, S, catS]$:26 |
| M13 | $[O, P, catS, S, cat]$:30 | $[O, S, catS, P, cat]$:24 | $[O, P, cat, S, catS]$:21 |
| M14 | $[O, S, catS, P, cat]$:23 | $[O, P, cat, S, catS]$:22 | $[O, P, catS, S, cat]$:22 |
| M15 | $[O, O, P, catS, S, cat]$:16 | $[O, O, P, catS, S, cat]$:15 | $[O, O, S, catS, P, cat]$:11 |
| M16 | $[O, P, catS, S, cat]$:31 | $[O, S, catS, P, cat]$:25 | $[P, O, cat, S, catS]$:22 |
| M17 | $[O, P, catS, S, cat]$:42 | $[O, P, cat, S, catS]$:28 | $[P, O, cat, S, catS]$:24 |
| M18 | $[O, P, catS, S, cat]$:35 | $[O, catS, S, P, cat]$:20 | $[O, P, cat, S, catS]$:18 |
| M19 | $[O, P, catS, S, cat]$:32 | $[O, P, S, cat, cats]$:30 | $[O, P, cat, S, catS]$:28 |
| M20 | $[O, O, S, catS, P, cat]$:9 | $[O, O, P, catS, S, cat]$:9 | $[O, O, S, P, cat, catS]$:8 |

