# OpenReview forum: "KinFormer: Generalizable Dynamical Symbolic Regression for Catalytic Organic Reaction Kinetics"
_ICLR.cc/2025/Conference — ICLR 2025 Poster_

### Official Review · Reviewer_c4Q1 · 2024-11-01

**Soundness:** 3
**Presentation:** 2
**Contribution:** 2
**Rating:** 3
**Confidence:** 2

**Summary:**

The authors address the kinetic equation prediction (KEP) problem for discovering symbolic equations from temporal concentration profiles of species while satisfying mass-action law constraint. They propose to combine a Transformer encoder/decoder, as used by prior work, with a novel MCTS search strategy over the generation order of the ODE equations following a conditional training strategy. They show the approach is able to generalize from the training data, out-perform two classical symbolic regression methods, and generalize to new reaction types. They provide in-depth analysis of generalization across various reaction mechanisms.

**Strengths:**

- The paper addresses an essential task for understanding chemical reactions.
- The methodology and analysis sections show a deep understanding of the problem.
- The generalization favors traditional SR methods

**Weaknesses:**

- There needs to be more baselines than traditional SR approaches and the three training strategies (which is more of an ablation)
- The topic does not seem very relevant for ICLR
- The approach does not seem scalable or robust to noise

**Questions:**

- How much does performance drop if you take random/bad generation orders for the conditional strategy?
- How can the robustness to noise be improved?

---

### Official Review · Reviewer_tWZW · 2024-11-03

**Soundness:** 3
**Presentation:** 3
**Contribution:** 3
**Rating:** 8
**Confidence:** 2

**Summary:**

# Modelling reaction kinetics with (language model based) dynamic symbolic regression and Monte-Carlo tree search

This paper presents a method to model reaction kinetics for a reaction between a given number of reaction partners (substrates, catalysts, products). In particular, the proposed method infers a symbolic form of the reaction mechanisms (as system of ODEs, in accordance with the mass action law), from given measurements of the concentration of each reaction partner, measured at several discrete timepoints along.

Compared to previous approaches, the current method does not generate all equations governing the simultaneously (referred to as *universal strategy* by the authors) or sequentially and independently (referred to as *independent strategy*). Instead, the authors propose to generate the order of the equations in the system of ODEs sequentially, but whilst conditioning subsequent equations on the previously generated equations within one prediction (ie. one system of ODEs). The prediction order is determined via Monte Carlo tree search.

The authors test their ideas on a set of 20 reaction mechanisms, from which synthetic temporal concentration profiles are simulated and then used as training and inference input data.

The contributions are:
1. The proposed strategy (in particular the conditioning of subsequent ODEs on previous ODEs within one reaction mechanism - ie ODE system and the MCTS post-processing to search for the optimal generation order).
2. The performance analysis and comparison of in-distribution, out of distribution (but within a given reaction class) and out of distribution for a different reaction class generalization. Here the authors compare to two prior methods and find significant improvements in generalization.

**Strengths:**

Disclaimer: I am not directly researching in this field and therefore not familiar with the latest literature.

As someone familiar with the overall research area, but unfamiliar with this task and formulation -- and the task specific literature -- I found this paper a refreshing read with interesting ideas and discussions.

The problem description and statement were very clear and well readable, and the method was also well explained.

I found the study with the in-distribution, out of distribution (but within reaction class) and across reaction classes well designed and interesting.

Based on what the authors describe, the proposed conditioning scheme & combination with MCTS appears novel to me.

**Weaknesses:**

While the description of the problem and the method were great, I found the discussion of the results comparatively harder to follow. This may be due to me coming from an adjacent field, but I would appreciate if the authors could refine this discussion further (c.f. my questions below for guidance).

**Questions:**

1. I did not quite understand to what extend MCTS is a post-processing procedure? Is it not also used during training in order to come up with the one desired prediction order in the first place, since that order is needed for the conditioning you are doing of one equation by the next?

2. On page 6, by dimension, presumably you mean the number of reaction partners? If so, it might help readers from a loosely adjacent field (like myself) to state this explicitly.

3. For your comparison against beam search, would you be able to go into a bit more detail on what your n=20 is here?

4. Figure 6: I'm afraid I cannot follow what this is meant to represent. Is the left hand side concentration profile the input you got from the timepoints of the exact simulation of your synthetic system (on the RHS)? And are the dotted lines the inferred symbolic estimates?


Very minor suggestion: There are various articles missing or at wrong places. I would kindly suggest the authors to run the text through a grammar checker to enhance the reading flow. Other than that it is very readable!

---

### Official Review · Reviewer_vrt5 · 2024-11-04

**Soundness:** 3
**Presentation:** 3
**Contribution:** 2
**Rating:** 5
**Confidence:** 3

**Summary:**

This paper develops dynamic SR for catalytic organic reaction kinetics from experimental time series, using a transformer approach.

**Strengths:**

The approach is reasonable. The paper seems well-written and well-presented. The appendix is detailed.

**Weaknesses:**

Overall, I think the paper has not sufficiently presented a convincing case of innovation and novelty. This may partly be due to the fact that I found the paper hard to understand. For example, in Figure 1a, the inner arrows is not explained in the caption, and I could not find it in the text. In Figure 1b, the constants with negative subscripts such as $k_{-1}$ are not explained.

The paper seems to define generalizability in the intro as being not domain-specific, but as far as I can tell, they then go on to only show results for a very narrow domain, but only for inter/intra class.

My understanding is that the physical constraints R are the mass-action law. I think a sentence or two more on how they maintain that would be helpful. And doesn't that then also preclude generalizability? How would they support different physical constraints (if they claim generalizability)?

Does the numerical tokenization scheme represent, say, the difference between 1.23 and 1.24, as being less than the difference between 4.21 and 8.21? How does it compare to things like XVal [1]?

I think my main issue with the paper is the significance of it. I would imagine that collecting the experimental data is often the largest effort. Once collected, is it difficult for a human to look at the data and fit the kinetics?

If this technique could be generalized to general ODEs, then I think it would be of significantly greater interest.

[1] Golkar, Siavash, et al. "xval: A continuous number encoding for large language models." arXiv preprint arXiv:2310.02989 (2023).

**Questions:**

Can this technique be useful in other kinds ODEs and/or PDEs?

Considering that this is *symbolic* regression, what does an RMSE error mean? For example, if it should be a +, but is instead a *, what is the RMSE?

---

### Official Review · Reviewer_5fkZ · 2024-11-09

**Soundness:** 3
**Presentation:** 3
**Contribution:** 3
**Rating:** 8
**Confidence:** 3

**Summary:**

This work presents a novel approach to solving dynamical symbolic regression through deep learning, with a specific emphasis on the dynamics of chemical reactions and kinetic equations. The central concept involves parameterizing the underlying ordinary differential equations (ODEs) using a conditional transformer that respects physical constraints aligned with the reaction dynamics. While this modeling technique performs effectively within the training distribution, it struggles with out-of-distribution scenarios.
To address this limitation, the authors introduce a conditional Monte Carlo Tree Search (MCTS) algorithm that seeks compatible solutions while adhering to the specified constraints. The proposed model, KinFormer, is evaluated across 20 symbolic regression benchmarks in organic chemistry, demonstrating competitive performance within the training distribution and significantly outperforming baseline models in out-of-distribution cases.

**Strengths:**

The paper is well-structured and enhanced by effective visualizations, aiding readers who may be familiar with ordinary differential equations (ODEs) and transformers, but not necessarily with reaction dynamics, in understanding the overall problem.

The methodology is interesting and clearly articulated. The central concept of parameterizing the dynamics using a conditional transformer, while ensuring that the model adheres to physical constraints, represents a promising research direction for integrating deep learning models into scientific and engineering contexts. Notably, although it has only been tested on relatively small benchmarks, the KinFormer framework seems to be scalable.

The approach of utilizing Monte Carlo Tree Search (MCTS) for out-of-distribution symbolic ODE generalization is both innovative and significant. Search strategies have proven essential in various fields, including discrete optimization, game theory, and reasoning within large language models. Leveraging search techniques is logical, provided that a consistent value/reward can be assigned to each node in the tree. While the underlying idea is straightforward, achieving effective implementation in such a technical domain is non trivial.
Although the focus is on chemistry, the potential for generalizing the use of transformers and MCTS extends to a wide range of scientific and engineering problems governed by ODEs.

**Weaknesses:**

### Node Scoring
The reasoning behind the scoring mechanism presented in subsection 4.2 lacks clarity. It would be beneficial to provide some intuitive insights into why the heuristic employed for Go is also effective in the context of symbolic regression for chemistry. This could enhance the reader's understanding and strengthen the argument.

### Limited Evaluation
The KinFormer has only been tested on 20 organic reactions that share similar properties. It would be valuable to conduct additional experiments across a broader range of fields, including inorganic chemistry. This would provide a more comprehensive assessment of the method's applicability and robustness.

### Cost
In Table 1, the complete KinFormer method demonstrates strong performance out-of-distribution (OOD); however, it is also 10 to 20 times more resource-intensive compared to the baseline methods. A deeper analysis of this trade-off between performance and computational cost would be insightful for readers considering its practical implementation.

**Questions:**

- How did you collect the data used in your research? Are all the reactions you analyze simulated numerically? Why did you focus only on organic reactions?

- Why did you choose to employ "prefix notation" for symbolic regression in your work?

- Is there something unique about the kinetic equation or reaction dynamics that enhances the effectiveness of your method?

- On average, how many simulation steps are necessary to derive the equations for the 20 organic reactions? Specifically, what is the average depth of the MCTS you observe? Is it essential to utilize all 100 steps in your simulations?

- Have you considered implementing a hybrid approach, such as integrating a NeuralODE or a numerical simulator during the MCTS?

---

### Meta-Review · Area_Chair_3ZXH · 2024-12-19

**Metareview:**

This paper applies neural ODEs to modelling kinetic equations. The paper is well-carried out clearly demonstrating domain expertise.

The reviews are bimodal with two reviewers strongly in favor of acceptance and two other reviewers clearly more reserved (with only one of them begin active in the discussion phase). Having gone over the paper, it is recommended that the paper is accepted.

**Additional Comments On Reviewer Discussion:**

None.

---

### Decision · Program_Chairs · 2025-01-22

Accept (Poster)